# Coupled Modal Analysis and Aerodynamics of Rotating Composite Beam

**DOI:** 10.3390/ma16237356

**Published:** 2023-11-26

**Authors:** Grzegorz Stachyra, Lukasz Kloda, Zofia Szmit

**Affiliations:** Department of Applied Mechanics, Faculty of Mechanical Engineering, Lublin University of Technology, 20-618 Lublin, Poland

**Keywords:** composite material, rotating structure, modal interactions

## Abstract

This study primarily focuses on conducting, both experimentally and numerically, a modal analysis of a cantilever composite beam. Through extended numerical simulations, we investigate Campbell diagrams, which, depending on the rotation speed of the structure, comprise natural frequencies and their corresponding modal shapes. Our results are categorized into two main aspects: the classical single-mode behavior and an innovative extension involving linearly coupled modal analysis. One key novelty of our research lies in the introduction of an analytical description for coupled mode shapes, which encompass various deformations, including bending, longitudinal deformations, and twisting. The most pronounced activation of dynamic couplings within the linear regime for a $45^{\circ }$ preset angle is observed, though the same is not true of the $0^{\circ }$ and $90^{\circ }$ preset angles, for which these couplings are not visible. In addition to the modal analysis, our secondary goal is to assess the lift, drag forces, and moment characteristics of a rectangular profile in uniform flow. We provide insights into both the static and dynamic aerodynamic responses experienced by the beam within an operational frequency spectrum. This study contributes to a deeper understanding of the dynamics of composite rotating beams and their aerodynamic characteristics.

## 1. Introduction

Beams are among the most popular construction elements in engineering. Therefore, it is crucial to have a solid understanding of both the basic and advanced theories related to beam modeling. This knowledge is not only applied but also enhanced in the case of rotating beams, which find widespread use in various industrial applications. Some of the most common applications include wind turbines, helicopter rotors, and airplane propellers.

In their work [1], the authors provided a comprehensive review of the most common theories pertaining to beams, which have been utilized by scientists over the past few decades. They examined classical approaches, such as those by Da Vinci, Euler-Bernoulli, and Timoshenko, in addition to the Generalized Beam Theory. Special emphasis was placed on the Carrera Unified Formulation (CUF) in one dimension, and the authors presented numerical examples illustrating its application in static, dynamic, and aeroelastic problems. Furthermore, they conducted an overview of two recently developed methods: axiomatic/asymptotic and component-wise approaches. The primary conclusion drawn from this critical review is that beam theories are still in need of further development and improvement. In another study by Wang et al. [2], a reduced model for vortex-induced vibrations (VIVs) in turbine blades is derived. In this study, the authors modeled the blades as uniform cantilever beams and employed the multiple scale method to investigate nonlinear dynamics. Subsequently, they calculated frequency–response curves and identified two types of bifurcation. The results presented underscored the necessity of employing coupled models to analyze the rich dynamics of VIVs.

The asymptotic development method is employed in [3] to investigate the free nonlinear oscillations of initially straight Timoshenko beams. The authors focused on two different definitions of curvature: one with respect to the deformed length and the other with respect to the undeformed length. The comparison of these two methods was the primary objective of their study, and the authors demonstrated that the results for slender beams are very similar when using both approaches. Furthermore, in [4], the authors analyzed the model of a geometrically exact nonlinear Timoshenko beam. They derived the equations of motion for the structure but primarily concentrated on one-dimensional constitutive equations. The paper presents basic numerical results. A similar approach is applied in [5] to analyze the dynamics of an elastic isotropic rotating beam. The eigenvalues and mode shapes are obtained for the linear problem, and the coupling between flapping, lagging, axial, and torsional components is studied. In the second part of the paper [6], the authors focused on analytical calculations. They applied the multiscale method directly to the partial differential equations of motion and drew backbone curves. Additionally, they analyzed three flapping modes as the angular speed varied from low to high. The scientists demonstrated that the nonlinearities of the flapping modes are strongly correlated with angular speed and can transition from hardening to softening and vice versa. In addition, Thomas et al. [7] conducted a study on the influence of rotation speed on the nonlinear vibrations of a cantilever beam. They focused on the phenomena of hardening/softening and jump effects, particularly when dealing with large amplitudes. To analyze these phenomena precisely, they employed three different models: two analytical models and one original model based on finite-element discretization. On a related note, the nonlinear vibrations of a rotating Timoshenko beam were investigated using the p-version finite element (FE) method in [8]. This study considered two types of nonlinearities: the strain–displacement relationship and the inertia force resulting from the rotation speed. Nonlinear forced vibrations were analyzed in the time domain, with consideration for both constant and non-constant rotation speeds. Carrera et al. [9] examined the free vibrations of a rotating composite blade. They employed the Carrera Unified Formulation (CUF) and the FE method to solve the governing equations. The authors placed their focus on both flapwise and lagwise motion, and they also accounted for the Coriolis force in their analysis. In a related study, presented in [10], the authors delved into the nonlinear vibration of a rotating beam with variable angular velocity. They concentrated on the coupling between longitudinal and bending vibrations. The authors derived the governing equations of motion using Hamilton’s principle and the Galerkin method. They then applied the multiscale method to obtain a first-order approximate solution. Their results were compared to those obtained through numerical integration, demonstrating a very good agreement. In the work of [11], the same methods were applied to derive the equations of motion for a rotating composite Timoshenko beam with both open and closed box-beam cross-sections. The authors stated that the change in pitch angle significantly influences the coupling between flapwise bending and chordwise bending motions, which is associated with the centrifugal force. The presented results take into account nonconstant angular speed as well as a nonzero pitch angle. Given the practical applications of rotating beams and structures, it is of paramount importance to consider the significant impact of aerodynamic loads on their dynamics. In the paper by DiNino et al. [12], an in-depth analysis of a homogeneous viscoelastic beam was conducted under the influence of uniformly distributed turbulent wind flow. This study encompasses an examination of both the steady and turbulent components of the wind, with a particular focus on their roles in Hopf bifurcation and parametric excitation. The authors also emphasized the interaction between bifurcation phenomena and the critical and post-critical behavior of the beam. Meanwhile, in [13], Elmiligui et al. present results obtained from numerical simulations of flow past a circular cylinder. Two distinct approaches are employed to prepare the model for simulations, and the resulting data are compared with previously published experimental findings. Nonlinear vibrations of the blade under high-temperature supersonic gas flow and varying angular speed are presented in [14]. The authors assume that the blade is pre-twisted, presetting, and a thin-walled rotating cantilever beam is used. The equations of motion are derived using Hamilton’s principle and the Galerkin method, revealing the presence of 1:1 internal resonance as well as primary resonance. The numerical results presented in the paper show that not only periodic motions but also chaotic motions can occur in the nonlinear vibrations of the rotating blade when the angular speed varies. Furthermore, in [15], a bifurcation analysis of a rotating pre-twisted beam is presented, taking into account varying speed and aerodynamic forces. The model is analyzed in both the chordwise and flapwise directions, revealing phenomena such as jumps, saturation, and double jumps. Additionally, in [16], a model of an Euler–Bernoulli beam with nonlinear curvature and coupled transversal–longitudinal deformation is introduced. The authors applied Hamilton’s principle to derive the equations of motion, with a focus on time delay control as the primary task. They presented the influence of linear and cubic control methods on vibration reduction for different rotating speeds. Meanwhile, nonlinear vibrations of a slowly rotating beam with a tip mass are studied in [17]. The authors applied the extended Euler–Bernoulli theory to analyze longitudinal–bending–twisting vibrations. They utilized the multiple time-scale method to solve partial differential equations and demonstrate the influence of angular speed, tip mass, and hub on nonlinear vibrations. Furthermore, the free vibrations of the beam model with a tip mass are explored in [18]. The authors focused on cross-sectional rotations, lateral bending, and transverse bending. The numerical simulations illustrate the effects of tip mass, rotary inertia, viscoelastic damping, and the beam inertia ratio on the stability of the system, as well as on natural frequencies. In their work, Huang et al. [19] presented fascinating results from experimental studies on slowly rotating cantilever beams. They employed Digital Image Correlation, the Phase Mapping Method, and direct measurements under operational conditions to analyze three beams, subjecting them to twenty different angular velocities. Their findings revealed centrifugal hardening behaviors in the flap-wise direction, confirming the accuracy of their chosen model. Notably, they achieved excellent agreement between experimental data and numerical calculations for hardening frequency. Another study of rotating composite beams is discussed in the paper by Gawryluk et al. [20]. In this research, the authors assumed a constant angular velocity for the rotating beam and utilized a Macro Fiber Composite${}^{\mathrm{TM}}$ (MFC) actuator for excitation. They employed numerical solutions via the FE method, which were subsequently validated through experimental testing. Additionally, Rafiee et al. [21] provided a critical review of scientific papers focused on rotating beams. The authors examined various approaches to calculations, including analytical, semi-analytical, and numerical methods, and discussed different beam theories. This paper offers a comprehensive overview of research on beam vibrations that has been conducted in recent years.

In a study by Teter et al. [22], modal analysis of a rotor composed of three active composite beams is presented. They compare experimental results obtained from a laser vibrometer and a LMS Test.Lab analyzer${}^{®}$ with modal hammer to numerical simulations performed using Abaqus${}^{®}$ software. The authors achieved excellent agreement among all methods, not only for natural frequencies but also for mode shapes. In the subsequent paper authored by Mitura et al. [23], the dynamics analysis of the rotor operating at a constant angular velocity is presented. The authors employed a Digital Signal Processing (DSP) system to excite vibrations in the beams and control angular speed. The authors investigated the influence of the piezoelectric effect and the hub’s speed on the rotor’s dynamic behavior. An analysis of force vibrations in a mistuned three-bladed rotor is presented in Warminski et al. work [24]. They assumed that beam mistuning in the rotor results from manufacturing processes in composite production. The rotor was excited by harmonic torque, or by chaotic oscillations. This study revealed the localization phenomenon. Furthermore, the localization and synchronization in a rotor with three beams were studied by Szmit in [25]. The model was analyzed numerically based on equations of motion and through numerical simulations using Abaqus${}^{®}$ software. Additionally, the paper presented results from experimental studies, including natural and force vibrations. Finally, Szmit et al. [26] conducted fully experimental studies on a three-bladed rotor. They used high-speed cameras during constant angular speed rotation to analyze the aerodynamic loads at different preset angles. The results include polynomials describing aerodynamic loads based on camera images.

Despite the extensive literature on rotating beams’ vibrations, in which the single-mode linear behaviour of the eigenvalue problem is corrected through nonlinear effects, the mechanical coupling, which already occurs in the linear problem between two distinct orthogonal modes, appears to be overlooked in the analytical/numerical models. This provided motivation for conducting a numerical modal analysis within detailed inspection of interactions already in the the linear scope. Furthermore, the linear dynamics of beams is supplemented with aerodynamic characteristics that are closely dependent on the beam’s geometry. Research on this aspect is lacking in the majority of studies on rotating beams.

The paper is organized as follows. In Section 2 linear modal analysis of the rotating structure are presented. Graphs illustrating the change in natural frequencies and associated linear mode shapes with rotor rotational speed are discussed, and linear mode couplings of bending, longitudinal motion and torsion in the spatial coordinate system are explored. The aerodynamic characteristics of static and dynamic lift/drag forces, together with aerodynamic momentum, are investigated in Section 3. The article concludes with final remarks and a description of future scientific research directions in Section 4.

## 2. Dynamic Response of Composite Structure

Let us consider a composite beam attached to a rigid hub with a radius *R*; see Figure 1. The beam is made of highly elastic ThinPregTM 120EP-513/CF resin and M4JB-12000-50B (TORAY) carbon fibers. Moreover, a specific stacking sequence [0/−60/60/0/−60/$60_{3}$/$-60_{2}$/$0_{2}$/−60/$0_{2}$/$60_{2}$/−60] ensures isotropic properties of material in the linear elastic regime, as defined by Hook’s law [27]. Uniform distribution of the material along the specimen’s length L=595 mm and cross-sectional area b×h=35 mm × 0.9 mm is assumed [25]. The effective mechanical properties of the composite structure are gathered in Table 1. In Figure 1a, only one coordinate system exists that rotates with the rotating beam–hub structure. The *x*-axis aligns with the longitudinal axis of the undeformed beam, the *z*-axis coincides with the hubs’ rotation axis and the *y*-axis completes the right-handed Cartesian coordinate system. Additionally, in Figure 1b, an angle Θ is measured from xy-plane positively defined in accordance with the right-hand rule about the *x*-axis. A preset angle Θ can be varied from $0^{\circ }$ to $90^{\circ }$, and describes the orientation of the blade attached to the hub. The system rotates with a constant speed $\dot{\varphi }$. The hub’s mass moment of inertia is infinite; hence, the rotating imbalance and inertial coupling between successive beams are not taken into account. The target of this assumption is to eliminate additional interactions between consecutive beams. The attention is devoted only to the beam as a 3D continuous structure, which can be deformed out of plane (outplane bending *i*), in-plane (inplane bending *j*), along the main of the beams’ axis (longitudinal *k*) as well as twist (torsion *l*). Note that, since the xyz coordinate system is embedded in the rotating hub, the directions of introduced deformations *i* and *j* are not aligned with the xy and yz planes. Only longitudinal *k* and twisting *l* motions can be referenced relative to the *x*-axis.

Commercial Ansys${}^{®}$ software was used for all numerical simulations presented in the paper. In the first step, the natural frequencies and associated mode shapes were validated in accordance with analytical calculations, simulations of competitive commercial software [17,28], as well as experimental studies performed in the absence of rotation ($\dot{\varphi }=0$) e.g., by neglecting the centrifugal force [16]. Nevertheless, the experimental investigations were restricted solely to the first two modes of natural vibrations. This prompted the authors to explore higher frequencies of the system’s natural vibrations within the frequency range that aligns with forthcoming numerical analyses.

Experimental measurements were conducted in the laboratory of the Department of Applied Mechanics at the Lublin University of Technology. An advanced PSV 500 laser scanning vibrometer and an electromechanical exciter SmartShaker K2007E01 were used for the measurements [22]. The measurement system setup is illustrated in Figure 2. The experiment was conducted based on a periodic chirp excitation in the frequency range of 0–25,132.7 rad/s (0–4 kHz), with the excitation applied at the base of the beam using the head of the electromechanical shaker. The calibrated scanning head performed three measurements for each of the 385 predefined points. Fast Fourier Transform (FFT) was then applied to the recorded time-domain signals for each point to identify resonant peaks and their associated vibration modes. The results of the vibration tests are presented as the first twenty detected modes of vibrations shown in Figure 3, and the corresponding magnitude–frequency plots are presented in Figure 4.

It is worth noting at this stage that in the absence of angular velocity, despite the use of a composite structure and a very broad frequency spectrum, no dynamic couplings were observed in the linear range of the dynamic response. The detected natural frequencies align with the numerical calculations reported in the Section 2.1. The authors regret that, due to technical constraints, they were unable to perform modal analysis considering a rotating structure. Consequently, experimental measurements were complemented only with numerical simulations using the finite element method.

### 2.1. Campbell Diagram

In this Section, the rotating system is axially pre-stressed due to centrifugal forces and then subjected to linear modal analysis. The distribution of centrifugal forces depends on angular velocity $\dot{\varphi }$ as well as the dimensions of the beam and the radius of the hub. It interferes with inertia and stiffness matrices and has a significant impact on eigenvalue problems such as linear eigenfrequencies and associated modes shapes. Natural frequencies as a function of rotational speed, e.g., Campbell diagrams, for five preset angles are presented in Figure 5. The linear natural frequencies up to 3000 rad/s are reported, and extended analysis for higher-order modes with logarithmic scale on ordinate are gathered in Appendix A.

In the angular velocity absence $\dot{\varphi }=0$, the natural frequencies are the same regardless of the radius of the hub and the preset angle of the blade. When rotation is activated, the two boundary angles $\Theta =0^{\circ }$ and $\Theta =90^{\circ }$ represent the dynamics of clear single-modes of vibration. In Figure 5a,b, despite numerous intersections of natural frequencies, no couplings occur. It is very interesting that for a rotational speed of 260 rad/s, three curves intersect at 650 rad/s. In general, values of natural frequencies increase with increasing rotational speed, but the slope trends are different. In contrast to the other curves, only the first torsion mode at $\Theta =0^{\circ }$ has a constant value of natural frequency. In the scenario in which linear modal couplings occur, indications on Campbell charts are not reported because they cannot be assigned to the conventional vibration modes included in the legend of the graphic. The individual interpretation will be performed in Section 2.3. Therefore, for preset angle $\Theta =30^{\circ }$, $\Theta =45^{\circ }$ and $\Theta =60^{\circ }$ strong linear modal interactions are observed in Figure 5b–d. In the first and third cases, the 1st inplane bending mode is lacking only for 300 rad/s and 400 rad/s, respectively. The preset angle $\Theta =45^{\circ }$ seems to be the critical one, for the sake of only torsion modes, and the 1st outplane modes were matched in the studied angular speed interval. This means that in the linear range, there is already a strong coupling or multiple instabilities in simulations on the beam.

### 2.2. Linear Mode Shapes

Campbell diagrams display only natural frequencies and lack information about the deformation of the shape. Based on three selected angular speeds, the change in linear mode shapes necessitates a proper discussion on deflection half-waves and the modal nodes location. To facilitate the observed changes, selected higher-order modes are presented in Figure 6. The third out-of-plane bending mode is very susceptible to angular velocity in the range of up to 1000 rad/s. Firstly, the natural frequency varies from 295.591 rad/s to 3917.31 rad/s. Secondly, the two modal nodes shift at approximately 1.5% and 0.84% at 100 rad/s. The changes become more prominent at approximately 3.7% and 3.5% at 100 rad/s. The mentioned shifts in % refer to nodals’ displacements over the length of the beam in the free rest configuration. Standardization of the results to the maximum beam deflection also shows that with the increase in rotational speed, the amplitude of the first two half-waves decays with respect to the free end. Moreover, the deflection arrows of the half-waves are inclined to the right. The second flexural in-plane mode shape remains constant for increasing rotational speed, while its natural frequency changes from 3968.35 rad/s to 4569.89 rad/s, which provides an increase of about 15% (see Figure A1). Analogously to the bending mode in the susceptible direction, torsional vibrations display a shift in the modal node by approximately 0.5% and 5.9% for 100 rad/s and 1000 rad/s, respectively. The second analogy is the reduction in the maximum twist in the first half-wave. We note that standard linear mode shape projections ($\Phi_{i}$, $\Phi_{j}$, $\Phi_{k}$ and $\Phi_{l}$) consisting of amplitudes ($A_{1}$–$A_{4}$, $B_{1}$–$B_{4}$, $C_{1}$–$C_{2}$ and $D_{1}$–$D_{2}$) and characteristic coefficients ($\lambda_{1}$, $\lambda_{2}$, $\lambda_{3}$ and $\lambda_{4}$) can be described in the form
(1)$$\begin{array}{c} \Phi_{i}\left(x\right)=A_{1}\sin\left(\lambda_{1}x\right)+A_{2}\cos\left(\lambda_{1}x\right)+A_{3}\sinh\left(\lambda_{1}x\right)+A_{4}\cosh\left(\lambda_{1}x\right), \end{array}$$
(2)$$\begin{array}{c} \Phi_{j}\left(x\right)=B_{1}\sin\left(\lambda_{2}x\right)+B_{2}\cos\left(\lambda_{2}x\right)+B_{3}\sinh\left(\lambda_{2}x\right)+B_{4}\cosh\left(\lambda_{2}x\right), \end{array}$$
for outplane/inplate bedning, and
(3)$$\begin{array}{c} \Phi_{k}\left(x\right)=C_{1}\sin\left(\lambda_{3}x\right)+C_{2}\cos\left(\lambda_{3}x\right), \end{array}$$
(4)$$\begin{array}{c} \Phi_{l}\left(x\right)=D_{1}\sin\left(\lambda_{4}x\right)+D_{2}\cos\left(\lambda_{4}x\right), \end{array}$$
for longitudinal and twisting are sufficient. Hoverer, the indicated amplitudes and characteristic coefficients must satisfy the sclerotic boundary conditions at x=0 and rheonomic constraints x=L by balancing internal forces, centrifugal forces, Coriolis forces and inertia terms. To date, finding an analytical solution to such a complex problem remains challenging.

### 2.3. Linear Mode Couplings

In this section, we devote attention to more sophisticated mode shapes, which involves combining at least two linear *unidirectional* mode shapes [29]. The introduced i,j,k,l notations can be extended to combined mode shapes Φ(i,j,k,l) in the linear regime, e.g., the third inplane bending mode interacting with the first longitudinal mode and second twist mode can be classified as Φ(0,3,1,2). In proposed notation, the frequency dependence is omitted for simplicity. Referring to Campbell charts of Figure 5c, a set of the most interesting solutions of numerical simulations for preset angle $\Theta =45^{\circ }$ and $\dot{\varphi }=800$ rad/s is presented in Figure 7. This is a particularly complicated case, in which twin modes of vibrations Φ(4,2,0,0) for $\omega_{7}=4164.62$ rad/s and $\Phi {\left(4,2,0,0\right)}^{*}$ for $\omega_{9}=4662.44$ rad/s are obtained. Despite the fact that both consist of the fourth flexible mode and the second flexible mode with greater stiffness, their natural frequencies differ. Moreover, in Figure 7a the *i*-type mode is dominant, while in Figure 7b, the *j*-type mode is more exposed. Since the notation counts only the dominant modes, it is conventional to implement weights ($\bar{A}$, $\bar{B}$, $\bar{C}$ and $\bar{D}$) for each mode of vibration
(5)$$\begin{array}{c} \bar{\Phi }\left(t,x\right)=\sin\left(\omega_{n}t\right)\Phi \left(i,j,k,l\right)=\sin\omega_{n}t\left[\bar{A}\Phi_{i}\left(x\right)+\bar{B}\Phi_{j}\left(x\right)+\bar{C}\Phi_{k}\left(x\right)+\bar{D}\Phi_{l}\left(x\right)\right], \end{array}$$
(6)$$\begin{array}{c} \bar{A}+\bar{B}+\bar{C}+\bar{D}=1, \end{array}$$
where $\omega_{n}$ corresponds to the *n*th natural frequency of a given $\dot{\varphi }$, while *t* is the time.

Other detected modal couplings include the combination of the 9th outplane bending with the 3rd in plane bending and 1st longitudinal with 10th torsion, which are presented in Figure 7c and Figure 7d, respectively. We have observed that neither the first in-plane bending singular mode nor any coupled modes are detectable; therefore, this mode can be subtly incorporated into other coupled modes of vibration.

Extended results for fixed preset angle $\Theta =45^{\circ }$ and gradually varied rotational speed $\dot{\varphi }$ for 200 rad/s, 300 rad/s, 400 rad/s, 500 rad/s, 600 rad/s, 700 rad/s, 900 rad/s and 1000 rad/s are reported in Figure A2, Figure A3, Figure A4, Figure A5, Figure A6, Figure A7 and Figure A8. These results provide a solid basis for further analyses using analytical methods, indicating the level of complexity of the issue in 4D space and in the time/frequency domain.

## 3. Aerodynamic Simulations

### 3.1. Lift/Drag Forces and Momentum

In this section, we focus on the aerodynamic aspects of the 2D blade in the flow of uniform air. The rectangular cross-section of the beam is placed at a given preset angle, Θ. In the studied case of a non-deformable structure, the preset angle is consistent with the angle of attack. Geometric details of the Computational Fluid Dynamics (CFD) simulations are presented in Figure 8. During the simulations, aerodynamic forces were recorded over time. In post-processing, the maximum and minimum magnitudes, as well as the mean values, of steady-state motion time histories were grouped according to the angle of attack Θ and varying airflow conditions. The translational airflow to rotation of the beam–hub structure is converted as follows
(7)$$\begin{array}{c} \dot{\varphi }=\frac{v}{R+L} \end{array}$$
where *v* corresponds to the air flow velocity at the tip of the blade and R+L describe the distance between the main axis of rotation and the beam tip in a free and undeformed configuration, with dynamic and centrifugal forces disabled. This assumption will be utilized to simplify our analysis.

Figure 9 presents lift drag forces in the function of preset angle Θ and angular speed $\dot{\varphi }$. At zero angle of attack, no lift force was detected, and the drag forces were at their minimum compared to the entire chart. Moreover, in the steady-state flow of the considered velocities, oscillations did not occur. Increasing the angle of attack to $5^{\circ }$ resulted in a significant increase in lift force with only a slight increase in drag force. Further increasing the angle of attack to $15^{\circ }$ and $30^{\circ }$ led to significant air resistance with only a minor rise in lift force. Karman vortices and the associated oscillations of forces in time histories appeared at an angle of attack of $45^{\circ }$. For this angle of attack, the values of lift and drag forces were nearly equal to each other. Subsequent changes in the angle of attack to $75^{\circ }$ and $90^{\circ }$ resulted in a significant increase in oscillations with increased drag and decreased lift forces. Furthermore, at a preset angle of $90^{\circ }$, the lift force oscillated around zero while drag reached its maximum values.

The analogous chart depicting the values of the aerodynamic moment acting on the beam is presented in Figure 10. For high airflow velocities, negative values of the aerodynamic moment for angles of attack at $5^{\circ }$, $15^{\circ }$, $30^{\circ }$, and $45^{\circ }$ draw attention. The remaining three angles of attack either exhibit zero moment values for $0^{\circ }$ or symmetric oscillations around zero for $75^{\circ }$ and $90^{\circ }$. It is worth noting that slight oscillations also occur at $\Theta =45^{\circ }$ and v=100 m/s, but they diminish with increasing velocity.

The above-mentioned aerodynamic loads can induce quasi-static deformations of mechanical system or excite its vibrations near resonance frequencies. This provides the foundation for examining another crucial aspect; namely, the frequency windows that impact the sample, along with the measurement of their magnitudes. In essence, we are establishing the groundwork for a comprehensive analysis of how specific frequency ranges may affect the sample and the extent of their influence.

### 3.2. Frequency Spectra

The time histories were subjected to a Fast Fourier Transform (FFT) to determine the airflow frequencies. In many cases, the frequency–magnitude plots exhibited one or two peaks. In order to consolidate the results, Figure 11 depicts a bubble chart on the rotational speed vs. response frequency plane. The bubble sizes are normalized to the dominant value, corresponding to the highest indication, while the remaining values (if present) are proportionally smaller.

For rotational speeds below 5 rad/s, oscillations occur only at $30^{\circ }$, $45^{\circ }$, and $75^{\circ }$. Additionally, two harmonics are excited only for preset angle $30^{\circ }$ and $45^{\circ }$. For angular speeds between 5 rad/s and 15 rad/s, a zero-degree angle of attack are inactive. However, for rotational speeds exceeding 20 rad/s, the first indication is observed at 150 rad/s, and the second at 300 rad/s.

The arrangement of bubbles can be divided into two groups: the first group includes angles of attack of $30^{\circ }$ and $45^{\circ }$ with a nonlinear trend of decreasing frequency indication values, while the second group includes $75^{\circ }$ and $90^{\circ }$ degrees with a trend of increasing frequency with the rotational speed $\dot{\varphi }$. The angle of attack $5^{\circ }$ has only three data points and appears to exhibit a linear trend. All time histories containing two distinct indications show that the lower harmonic has a greater value. However, it is essential to consider both frequencies to excite or avoid vibrations when their values coincide with the natural vibration frequencies, as depicted in the Campbell diagrams in Figure 5 and Figure A1 as well as coupled vibration mode shapes presented in Figure 7, Figure A2, Figure A3, Figure A4, Figure A5, Figure A6, Figure A7 and Figure A8.

## 4. Final Remarks and Further Developments

The numerical tools presented in the paper for the FE method and CFD simulations depict the issues related to rotating composite laminates, in which, in addition to the specimen fabrication processes, rotational speed play an important role. After presenting the linear single modes of vibrations and their corresponding natural frequencies, the focus shifted towards linear modal interactions and their deformation field description using an analytical method, considering a combination of two bending directions, *i* and *j*; longitudinal motion *k*; and torsional *l* mode shapes. The mechanical system can be subjected to external loads arising from aerodynamic flow and centrifugal forces. Depending on whether we want to avoid vibrations or excite them, the natural frequencies must be either isolated from the excitation frequencies or targeted. Beyond the excitation frequency, the amplitude and the force/momentum distribution represent critical factors that have a direct impact on the efficiency of motion excitation. It is worth mentioning the possibility of indirectly exciting vibrations, for instance, by stimulating torsional modes and utilizing mechanical couplings to induce significant longitudinal motion. One should consider various types of internal resonances, external subharmonics, and superharmonics, which may arise from both linear and nonlinear mechanical couplings.

In the future research development of rotating composite structures, three main topics will be explored: (i) analytical modelling of vortex-induced vibrations, (ii) the utilization of an electromechanical system for energy harvesting from mechanical vibrations, and (iii) control of coupled vibrations via MFC patches. The first topic involves the expansion of an analytical model presented in the [17]. Besides the nonlinear beam model and aerodynamic flow, there is a plan to incorporate nonlinear Van der Pol equations based on the aerodynamic characteristics of the rectangular cross-section beam and its associated frequency spectra. The second research area includes experimental measurements on the prototype presented in [26]. Based on the vibration modes, it is possible to estimate the optimal location for a harvester that operates proportionally not to the maximum displacement amplitude, but to the maximum curvature. Furthermore, linearly coupled modes of vibrations appear to be a choice of higher efficiency. The final issue pertains to the vibration suppression during rotor operation, which involves avoiding resonant frequencies associated with aerodynamic flow or an vibration reduction active control by piezoelectric transducers.

## 5. Conclusions

The primary objective of this study was to perform experimental and numerical modal analysis of a composite cantilever beam. In the laboratory investigations with specialized vibration measurement equipment, detailed maps of beam deformations (mode shapes) and magnitude-frequency curves were executed. Additionally, in the absence of rotation, the study conclusively affirmed the absence of linear modal interactions in the composite beam. Through numerical simulations, we delved into the intricacies of Campbell diagrams, determining the natural frequencies and their corresponding modal shapes. Our findings were classified into two distinct categories: the classical single-mode behavior and the pioneering extension of linearly coupled modal analysis. The two fundamental preset angles of $\Theta =0^{\circ }$ and $\Theta =90^{\circ }$ did not have dynamic couplings within the linear range. However, when the preset angle was adjusted to $\Theta =30^{\circ }$ and $\Theta =60^{\circ }$, couplings between linear mode shapes in the plane emerged. Rotational speeds exceeding $\dot{\varphi }=300$ rad/s and a preset angle of $\Theta =45^{\circ }$ proved to be the most linearly coupled, with flexural–flexural and longitudinal–torsion modes strongly interfering with each other, respectively. The results for the preset angle were categorized based on the coupling type and presented graphically.

Notably, we introduced an analytical description of coupled mode shapes, encompassing various deformations such as bending, longitudinal deformations, and twisting. This contribution is a noteworthy advancement in understanding the behavior of rotating structures.

In addition to the modal analysis, our secondary objective was to assess the lift, drag forces, and moment characteristics of a rectangular profile in uniform flow. The preset angle variation between $\Theta =0^{\circ }$ and $\Theta =90^{\circ }$ demonstrated lift/drag force transmission, in which, for the preset angle of $\Theta =45^{\circ }$, these forces were almost equal over the entire range of rotational speeds. Comprehensive insights into both the static and dynamic aerodynamic responses acting upon the beam within its operational frequency spectrum were provided. For preset angles $30^{\circ }$ and $45^{\circ }$, the vortex-induced vibrations occurred at very small angular speed $\dot{\varphi }=1$ rad/s, exhibiting two prominent harmonics.

This study represents a substantial step forward in the field of composite rotating beams, offering a deeper understanding of their dynamic characteristics and their interaction with aerodynamic forces. These findings hold significant promise for various engineering applications and contribute to the broader knowledge of dynamic systems.

## Figures and Tables

**Figure 1 materials-16-07356-f001:**
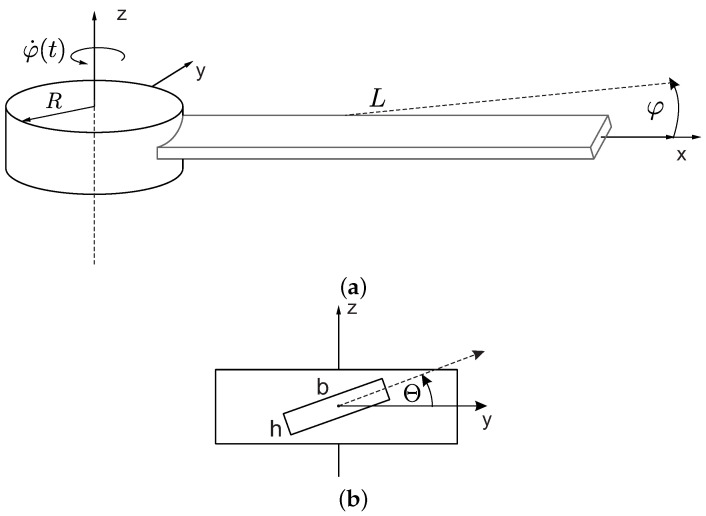
The beam–hub structure: (**a**) an isometric top view, and (**b**) a viewpoint orthogonal to the primary axis of the beam.

**Figure 2 materials-16-07356-f002:**
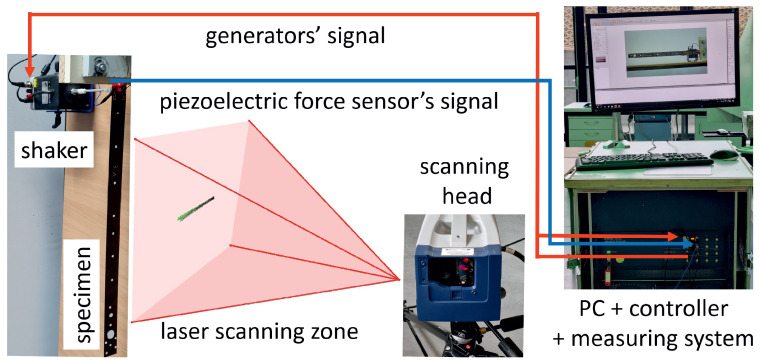
The scheme of the experimental setup.

**Figure 3 materials-16-07356-f003:**
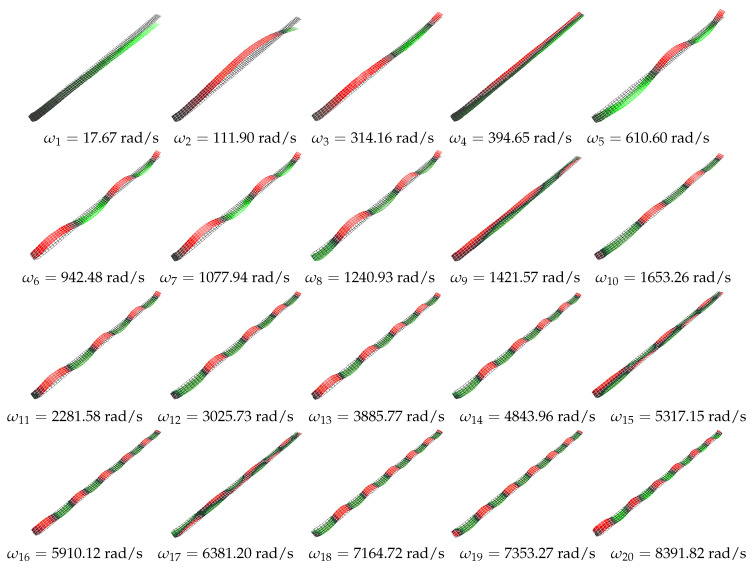
Experimental linear (single) mode shapes.

**Figure 4 materials-16-07356-f004:**
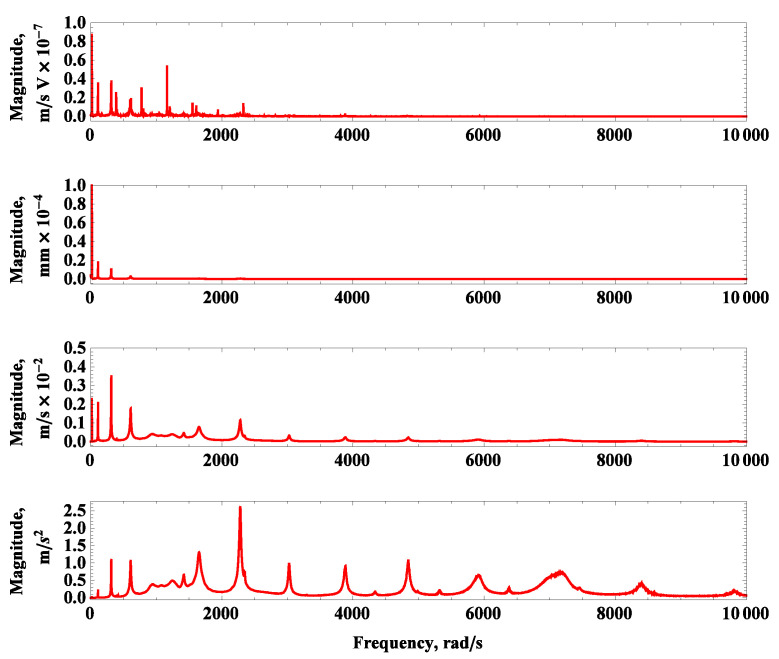
Frequency magnitude curves of dynamic tests for measured signals: velocity multiplied by force sensors’ voltage, displacement, velocity, and acceleration (from **top** to **bottom**).

**Figure 5 materials-16-07356-f005:**
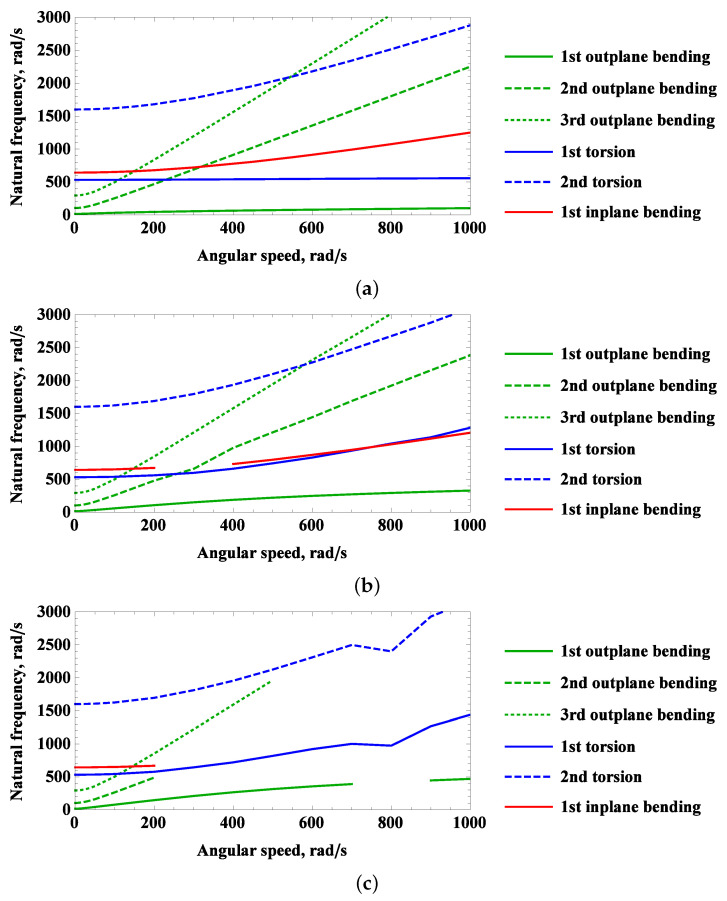
Campbell diagram of the rotating beam for predefined preset angle: (**a**) $\Theta =0^{\circ }$, (**b**) $\Theta =30^{\circ }$, (**c**) $\Theta =45^{\circ }$, (**d**) $\Theta =60^{\circ }$, (**e**) $\Theta =90^{\circ }$.

**Figure 6 materials-16-07356-f006:**
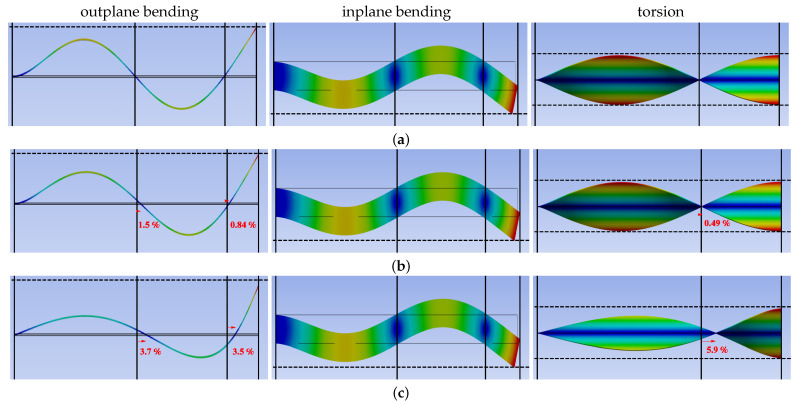
Linear mode shapes for preset angle $\Theta =90^{\circ }$ and varying angular velocity: (**a**) $\dot{\varphi }=0$ rad/s, (**b**) $\dot{\varphi }=100d$ rad/s and (**c**) $\dot{\varphi }=1000$ rad/s.

**Figure 7 materials-16-07356-f007:**
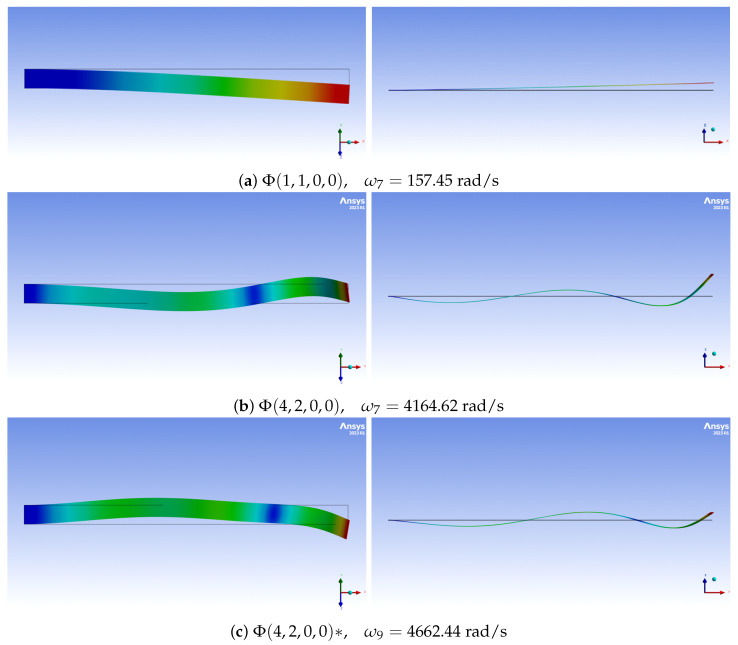
Linear mode couplings for preset angle $\Theta =45^{\circ }$ and $\dot{\varphi }=800$ rad/s.

**Figure 8 materials-16-07356-f008:**
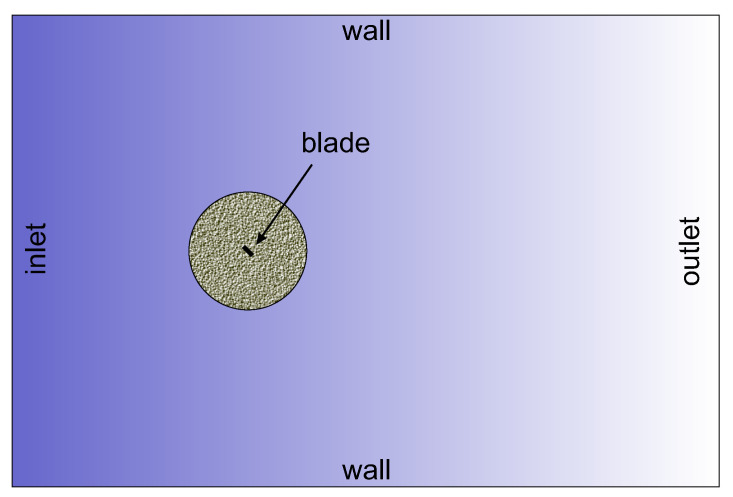
Geometry of CFD simulation domain 60D×40D (upstream 20D and wake 40D), the near-field cylinder 10D and the blade 1D corresponding to the width of the beam *b*.

**Figure 9 materials-16-07356-f009:**
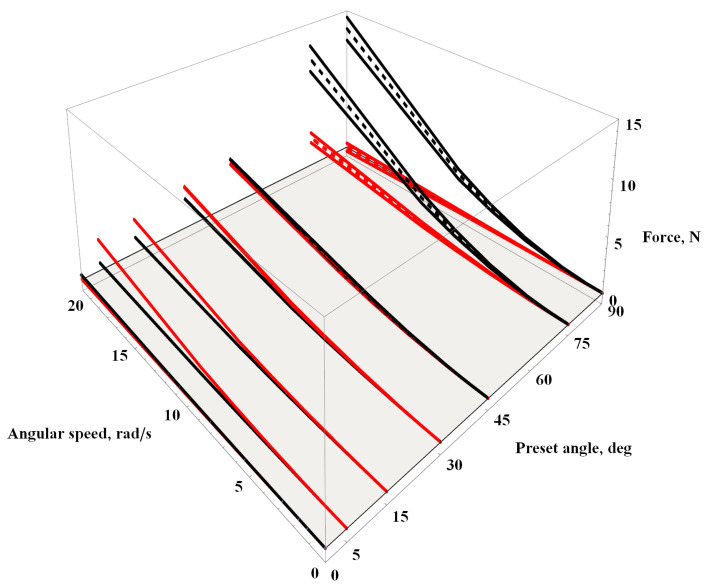
Aerodynamic lift (**red**) and drag (**black**) forces acting on the beam for the constant flow rate; see Equation (Equation 7).

**Figure 10 materials-16-07356-f010:**
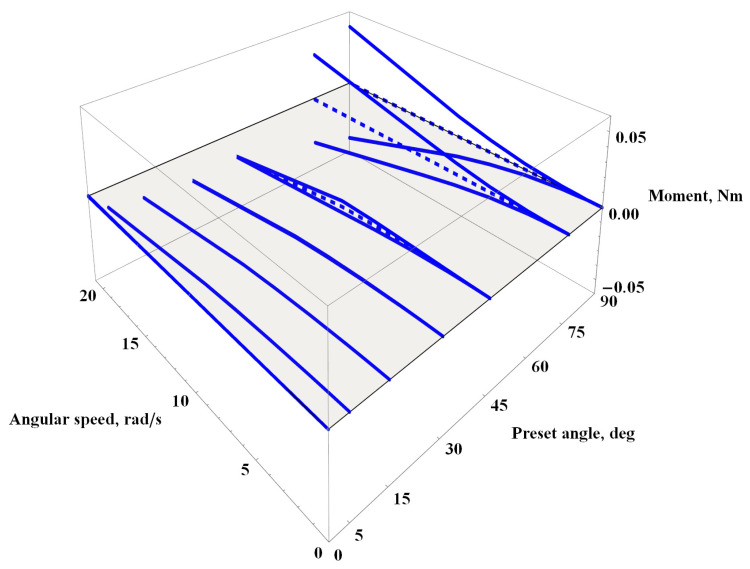
Aerodynamic momentum acting on the beam for the constant flow rate, see Equation (Equation 7).

**Figure 11 materials-16-07356-f011:**
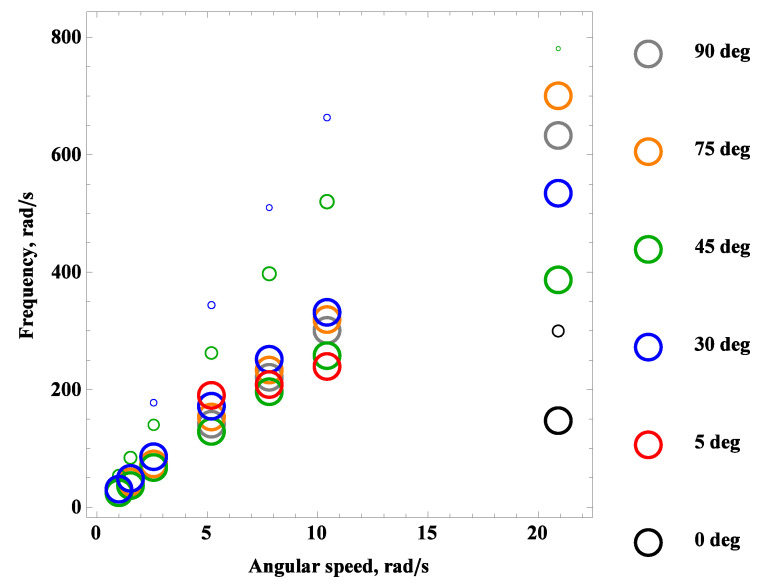
Frequency spectra of aerodynamic loads.

**Table 1 materials-16-07356-t001:** Effective mechanical properties of the composite beam: density, mass per unit length, Young’s modulus, shear modulus, Poisson’s ratio [16].

ρ	μ	*E*	*G*	ν
**kg/m^3^**	**kg/m**	**GPa**	**GPa**	**(-)**
1350	0.042525	55.7225	20.4862	0.36

## Data Availability

The data presented in this study are available on request from the corresponding author.

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
