# Peer review of "Coupled Modal Analysis and Aerodynamics of Rotating Composite Beam"

_materials, 2023, doi:10.3390/ma16237356_

Round 1

Reviewer 1 Report

Comments and Suggestions for Authors

The authors present a numerical study where they perform a modal analysis of a composite rotating beam and assess the natural frequencies and associated modal linear shapes. The reviewer main concern is about the novelty of the work. What are the reasons/motivations to perform this study. The paper is very well written and is easy to follow. The content of research is a valid contribution for the scientific community. The Reviewer recommends the paper for publication with major revision. There are some major issues that should be attended before accepting the paper for publication.

MAIN ISSUES TO BE ADDRESSED

The number of referenced 7 in the introduction section should be placed near the authors name. This is: Thomas et al. [7]...

The same issue should be amend in Reference 9. Please place the reference number near the author name.

In line 86 is pretwisted and in line 91 is pre-twisted. If the reviewer is correct the the correct word is pre-twisted. Please amend this.

In line 115, what is the meaning of the MFC abbreviation?

In line 124, what is the meaning of LMS abbreviation?

Line 125: Abaqus is a commercial name, therefore it need the registered trademark symbol (R).

In line 128, what is the meaning of the DSP abbreviation?

In line 136 again abaqus needs the registered symbol.

In the introduction section is missing a final paragraph stating the main objectives of the work and the major findings. Please include this paragraph.

Section 2: If the reviewer is correct the authors state that the beam or blade has 16 layers and the thickness is only 0.9mm?  It gives a layer thickness of 0.0565mm... Is this correct?

Line 164: where is the location of the twist axis l?

Table 1: only this properties? This a composite it needs the properties in all the three directions.

line 165: Ansys is a commercial name. It needs the registered symbol.

In figure 2, what is the reason for the curves not be completed. Specially the red curves (1st inplane bending) in almost all the graphs.

Line273 and 274. The conclusion is not in the opposite. This is in the graph is the drag force that increases.

What are the conclusions of the present work? What is the reason for the majority of the conclusions section being related to future works?

Reviewer 2 Report

Comments and Suggestions for Authors

The article presents simulation only. Why did not the authors validate their model in advance, and compare it to the experimental data? Linear mode shape has been presented here but the experimental validation is important to be examined. In addition, the aerodynamic simulation needs experimental validation. It is very hard to believe the simulation shows the real physics phenomenon in this particular case. This article is not recommended to be published in this Journal. 

Reviewer 3 Report

Comments and Suggestions for Authors

In the paper “Coupled modal analysis and aerodynamics of rotating composite beam,” written by Grzegorz Stachyra, Lukasz Kloda, and Zofia Szmit, modal analysis of a composite rotating beam is performed via Campbell diagrams, which consist of natural frequencies and associated modal linear shapes.

There are several concerns that make the reviewer disagree with the publication in its current form. Here is why:

1. The abstract needs to be largely revised. The current abstract only introduces the goals of the study which is not enough. The following things need to be addressed in the abstract: A brief introduction to the topic, an explanation of why the topic is important in your field, a statement about what the gap is in the research, your research questions/aims, an indication of your research methods and approach, a summary of your key findings, an explanation of why your findings and key message contribute to the fields.

2. Introduction also needs to be largely revised. There are 28 references in total in this paper and 26 of them are just listed with simple explanations of each work. The authors should address what is lacking in these papers and why the proposed work is necessary to overcome those drawbacks. The novelty has to be clearly identified in the introduction.

3. Line 151. The authors specify the beam used as ThinPregTM 120EP-513/CF resin and M4JB-12000-50B (TORAY) carbon fiber, however, this is not necessary for this work since no experiments were performed and the authors already gave material properties in Table 1. By the way, where do the values in Table 1 come from? Put the reference.

4. Typo in line 154. definied -> defined.

5. It is mentioned in line 166 that the natural frequencies and associated mode shapes were validated using analytical calculations however the reviewer can’t find any validation work afterwards.

6. Figure 2. Why are there discontinuities in (c) and (d) at angular speed of around 700~800 rad/s? 

7. What is the point of putting the figures of Appendix A? They are just listed without further discussion. 

8. Typo in line 253. coross -> cross

9. Overall, this manuscript is like a technical report or case study rather than a journal paper. The reviewer hardly sees the novelty. 

Comments on the Quality of English Language

A minor spell check is required.

Round 2

Reviewer 1 Report

Comments and Suggestions for Authors

All the questions posed by the reviewer have been duly answered by the authors in the revised version of the manuscript. The Reviewer therefore recommends acceptance of this manuscript.

Author Response

We would like to express our sincere gratitude for the time and effort you invested in preparation of a constructive review of our article. Your attention and thorough assessment significantly contributed to improving the quality of our work. It is thanks to your valuable insights and suggestions that we were able to enhance the content and increase the value of our article.

Reviewer 2 Report

Comments and Suggestions for Authors

1.     The abstract is missing key results. What are your specific and quantitative results? What could we identify from your study?

2.     You need a clear definition symbol for Table 1. What is rho, μ, E, G, v?

3.     What version of ANSYS®?

4.     It is unclear how to find Amplitudes and Coefficients for Eq. 1 – Eq. 4.

5.     Similarly, how to determine the weight (A,B,C,D) in Eq 5 and 6? I pointed out to change the symbol. They are the same as the amplitude symbol in Eq. 1-4.

6.     Don’t you have experimental data to validate your modal simulation? At least there are experimental validations for modal.

7.     What are the quantitative results in the conclusion?

8.     The article needs nomenclature.    

Author Response

Authors would like to thank the Reviewer for constructive comments and suggestions which improve the quality of the paper. The manuscript has been revised to address all the Reviewers’ suggestions. Below the answers are given point-by-point while the essential changes in the manuscript are introduced in a red color. 

  • The abstract is missing key results. What are your specific and quantitative results? What could we identify from your study?

A new sentence in  introduction has been introduced in red.

  • You need a clear definition symbol for Table 1. What is rho, μE, G, v?

Caption of Table 1 has been elaborated to explain parameters.

  • What version of ANSYS®?

The newest version of ANSYS® (2023 R2) is used, however, to the best of the authors' knowledge, in older versions of this software, the same results can be achieved.

  • It is unclear how to find Amplitudes and Coefficients for Eq. 1 – Eq. 4.

Amplitudes and characteristic coefficients must satisfy boundary conditions, for more details we refer to the paper co-authored by the corresponding author [17]. At this point we emphasize that the novelty of the current work is the linear coupling of vibration modes, which was not taken into account in the latter paper. 

  • Similarly, how to determine the weight (A,B,C,D) in Eq 5 and 6? I pointed out to change the symbol. They are the same as the amplitude symbol in Eq. 1-4.

Firstly, authors disagree with the Reviewer, because above mentioned symbols with and without subscript can define different terms. Moreover, normalization of mode shapes,  e.g. maximum deflection of the shape to be equal 1 (A1-A4 and λ1 manipulation) and then the gain normalized shape by amplitude A  is very intuitive and one of the most common practices in analytical calculations. We admit that we propose an analytical description, but the balancing of coefficients has been left for future solutions, most likely related to the balance of mechanical energy in the system.

  • Don’t you have experimental data to validate your modal simulation? At least there are experimental validations for modal.

We took the liberty of including a copied response to one of the Reviewers (R1). This response reads as follows:

"In Table 2 of the paper Kloda, L.;Warminski, J. Nonlinear longitudinal–bending–twisting vibrations of extensible slowly rotating beam with tip mass. International Journal of Mechanical Sciences 2022, 220, 107153. doi:10.1016/j.ijmecsci.2022.107153 natural frequencies obtained from finite element simulations (done in Abaqus®) and analytical model for the first natural frequencies in the given directions are reported. These data correspond to the natural frequency values on the Campbell diagram for a zero rotational speed and demonstrate excellent agreement. 

In Figure 2 of the paper Warminski, J.; Kloda, L.; Lenci, S. Nonlinear vibrations of an extensional beam with tip mass in slewing motion. Meccanica 2020, 55, 2311–2335. doi:10.1007/s11012-020-01236-9 linear modal shapes for the out-plane oscillations as well as longitudinal direction (cases with no tip mass) are gathered. It was the second step of confirmation between analytical model and finite element studies in Ansys® software.  The higher order natural frequencies for susceptible to bending directions are presented in Appendix 1, Table 1 of the same paper. 

Finally, in Section 3 of paper Warminski, J.; Kloda, L.; Latalski, J.; Mitura, A.; Kowalczuk, M. Nonlinear vibrations and time delay control of an extensible slowly rotating beam. Nonlinear Dynamics 2021, 103, 3255–3281. doi:10.1007/s11071-020-06079-3, experimental results are reported. The comparison matches perfectly with finite element simulations of the present study."

For that reason, we have obtained robust validation of the modal analysis, as depicted in the second paragraph of Section 2, titled 'Dynamic Response of Composite Structure.

  • What are the quantitative results in the conclusion?

The conclusions section has been enriched with quantitative results.

  • The article needs nomenclature. 

Authors checked random five papers published in  Materials and none of them had nomenclature. Therefore, we do not want to go beyond the Journal's standards and have not created it.

Reviewer 3 Report

Comments and Suggestions for Authors

The manuscript has been sufficiently improved to warrant publication in Materials based on the reviewers' comments. It can be published in its current form.

Round 3

Reviewer 2 Report

Comments and Suggestions for Authors

The article revision is unsatisfactory.

1.     The amplitude and weight symbols are the same. I pointed out to revise the weight symbol. Not all readers understand your equations.

2.     The modal simulation must be validated experimentally. Your own experiments or previously published experiments in the same condition.

3.     The article needs nomenclature.   

Author Response

The most recent version of the article incorporates all three comments made by the Reviewer. This implies that the Authors made revisions or adjustments based on the feedback provided by the Reviewer, ensuring that each of the three comments has been addressed and integrated into the updated version of the document.

  • The amplitude and weight symbols are the same. I pointed out to revise the weight symbol. Not all readers understand your equations.

The weights have been altered to terms featuring overbars.

  • The modal simulation must be validated experimentally. Your own experiments or previously published experiments in the same condition.

In previous response to the Reviewers' comments, the selected numerical results were validated against literature. However, recognizing the steadfastness of the Reviewer, new, original vibrational tests were conducted and incorporated into the work in Section 2. Due to hardware and technological limitations, we had to confine ourselves to a system without rotation. These tests confirmed the absence of modal interactions in the case of fixed beam.

  • The article needs nomenclature.

The authors introduced a nomenclature at the outset of the article.

Round 4

Reviewer 2 Report

Comments and Suggestions for Authors

The article has been revised sufficiently according to this reviewer's comments. The article is acceptable for publication in the Journal.